



# Disentangling fast and slow responses of the East Asian summer monsoon to reflecting and absorbing aerosol forcings

Zhili Wang[1], Lei Lin[2], Meilin Yang[3], Yangyang Xu[4], Jiangnan Li[5]

[1]State Key Laboratory of Severe Weather and Key Laboratory of Atmospheric Chemistry of CMA, Chinese Academy of
Meteorological Sciences, Beijing, 100081, China
[2]School of Atmospheric Sciences and Guangdong Province Key Laboratory for Climate Change and Natural Disaster Studies,
Sun Yat-sen University, Zhuhai, 519000, China
[3]Institute of Urban Meteorology, China Meteorological Administration, Beijing, 100089, China
[4]Department of Atmospheric Sciences, Texas A&M University, College Station, Texas, 77843, USA
[5]Canadian Centre for Climate Modelling and Analysis, Science and Technology Branch, Environment Canada, Victoria,
V8P5C2, Canada

*Correspondence to*: Lei Lin (linlei3@mail.sysu.edu.cn)

**Abstract.** We examine the roles of fast and slow responses in shaping the total equilibrium response of the East Asian

summer monsoon (EASM) to reflecting (sulfate, $SO_4$) and absorbing (black carbon, BC) aerosol forcings over the industrial

era using the Community Earth System Model version 1. Our results show that there is a clear distinction between fast and

slow responses of the EASM to aerosol forcings and the slow climate response due to aerosol-induced change in sea surface

temperature plays an important role in the impacts of aerosols on the EASM. The EASM is weakened by a decrease in land-

sea surface thermal contrast in the fast response component to $SO_4$ forcing, whereas the weakening is more intensive by the

changes in tropospheric thermodynamic and dynamic structures in the slow response component to $SO_4$. The total climate

adjustment caused by $SO_4$ is a significant weakening of the EASM and a decrease in precipitation. The BC-induced fast

adjustment strengthens the EASM both by increasing the local surface land-sea thermal contrast and shifting the East Asian

subtropical jet northwards. BC-induced slow climate adjustment, however, weakens the EASM through altering the

atmospheric temperature and circulation. Consequently, the EASM is enhanced north of 30°N but slightly reduced south of

30°N in the total response to BC. The spatial patterns of precipitation change over East Asia due to BC are similar in total

response and slow response. This study highlights the importance of ocean response to aerosol forcings in driving the

changes of the EASM.

## 1 Introduction

The East Asian summer monsoon (EASM) is one of the most complex and influential monsoon systems over the globe

(Ding and Chan, 2005). The activities of about 20% of world's population would be affected by rainfall change due to the

variation of the EASM (Lei et al., 2011). Further understanding of the features of the EASM change has important





implications for social economics, agriculture, ecosystem, and water resource management (Hong and Kim, 2011; Auffhammer et al., 2012).

The long-term variation of the EASM is possibly attributed to the influence of various factors including natural factors (e.g., internal climate variability, volcanic eruptions, and solar variability) and anthropogenic factors (e.g., anthropogenic aerosols and greenhouse gases (GHGs)) (Wang et al., 2001; J. Li et al., 2010; Salzmann et al., 2014; Wang et al., 2015; Z. Li et al., 2016). Among them, aerosol forcing has been recognized as an important contributor to the long-term change. The analyses based on the Coupled Model Intercomparison Project phase 5 (CMIP5) multimodel simulations indicated that aerosol forcing contributed dominantly to the weakening of the Asian summer monsoon during the second half of the twentieth century (Salzmann et al., 2014; Song et al., 2014). Other previous studies based on individual climate models also showed that the increases in anthropogenic aerosols could decrease the land-sea surface thermal contrast, thereby leading to a weakening of the EASM (e.g., X. Liu et al., 2011; Zhang et al., 2012; Jiang et al., 2013; J. Liu et al., 2017; Wang et al., 2017).

Despite the modeling and observational evidence, there is still debate over whether the total aerosols enhance or weaken the EASM (Guo et al., 2013; Yan et al., 2015), which could be related to the complicated nature of aerosol chemical compositions, an issue we aim to address in this study. Aerosols in the atmosphere consist of optically reflecting and absorbing components. Reflecting aerosols (e.g., sulfate ($SO_4$) and organic carbon) can cool the surface by decreasing the amount of sunlight arriving at the top of the atmosphere (TOA) and surface, and cause weak cooling inside the atmosphere due to a weakened solar absorption (Myhre et al., 2013). However, absorbing aerosols (e.g., black carbon (BC), dust, and some component of organic carbon) are able to not only change the radiation budget at the TOA and surface but also directly heat the atmospheric column (Koch and Del Genio, 2010; Huang et al., 2014). Consequently, BC affects the atmospheric stability, cloud cover, and convection. Therefore, the impact of aerosols on climate derived from modelling studies is likely to be substantially different when various aerosol species are accounted for (Ocko et al., 2014). Using GISS model, Menon et al. (2002) suggested that the "wetter-south-dryer-north" phenomenon that has appeared frequently in summer over eastern China during the past decades may be related to the increase in BC emission. But Zhang et al. (2009) showed responses that are opposite to that in Menon et al. (2002) when considering the integrated effects of carbonaceous aerosols.

Several studies attempted to contrast the $SO_4$ and BC responses and indicated that scattering and absorbing aerosols would have markedly different effects on regional temperature, atmospheric circulation, and precipitation over East Asia (e.g., Guo et al., 2013; Jiang et al., 2013; Persad et al., 2014). However, these studies all only considered the fast adjustments of atmosphere and land surface to aerosol forcings, without considering the response of oceans. Climate response to a forcing agent can be regarded as a synthesis of fast and slow responses (Andrews et al., 2010; Ganguly et al., 2012). The response to direct effects of aerosols on radiation, cloud, atmospheric heating rate, and land surface is treated as the fast response, while the response to change in global surface temperature, especially sea surface temperature (SST), caused by the aerosol forcing is identified as the slow response. The latter can have a more important effect on the climate system (Allen and Sherwood,



2010; Ganguly et al., 2012; Xu and Xie, 2015; Voigt et al., 2017). A general circulation model study by Hsieh et al. (2013) showed that aerosols could lead to different spatial responses of climate over the global scale when using an interactive ocean model as opposed to fixed SST as the ocean boundary conditions. Ganguly et al. (2012) also indicated that the slow component played a more critical role in shaping the total equilibrium response of the South Asian summer monsoon to
aerosol forcing.

The East Asian monsoon is considered as a more complex monsoon system. What role does the feedback of oceans to aerosol forcings play in driving the changes of the EASM? This study explores the roles of fast and slow responses in forming the total equilibrium response of the EASM to both reflecting and absorbing aerosol forcings over the industrial era using a state-of-the-art Earth system model. We take $SO_4$ and BC as the representatives of reflecting and absorbing aerosols
separately. To our knowledge, no previous study has partitioned the fast and slow responses of the East Asian monsoon to various aerosol species using a fully coupled climate model.

The paper is organized as follows. The model and simulations performed are described in Sect. 2. The total, fast, and slow responses of the EASM to various aerosol forcings are presented in Sect. 3. Our discussion and conclusions are summarized in Sect. 4. We primarily focus on the variation of the EASM over the region 20°-40°N, 100°-140°E. The summer includes
the months of June, July, and August (JJA).

## 2 Method

### 2.1 Global climate model

We used the Community Earth System Model version 1 (CESM1), a fully coupled ocean–atmosphere–land–sea-ice model, created by the National Center for Atmospheric Research (NCAR) of the U.S. (Hurrell et al., 2013). The model is a version
with a finite-volume approximate 1° horizontal resolution (latitude 0.9° × longitude 1.25° for the atmosphere and land, and 1° × 1° for the ocean) and 30-level vertical resolution, with a rigid lid at 4 hPa. CESM1 includes the primary anthropogenic forcing agents, such as GHGs, tropospheric and stratospheric ozone, sulfate, black, and primary organic carbon. The three-mode modal aerosol model that contains Aitken, accumulation, and coarse modes has been implemented in the model (Liu et al., 2012). It can provide the number and mass concentrations of internally mixed aerosols for the three modes. The model
also includes the physical representations of aerosol direct, semi-direct, and indirect effects for both liquid and ice phase clouds (Morrison and Gettelman, 2008; Gettelman et al., 2010; Ghan et al., 2012).

Anthropogenic and biomass burning emissions of aerosols and their precursors are based on Lamarque et al. (2010). However, the BC emission at present day is adjusted due to the potential underestimation of BC heating in the atmosphere in CESM1 (Xu et al., 2013; Xu and Xie, 2015). BC emissions over East Asia and South Asia are increased by a factor of 2 and
4, respectively. The emissions are changed in all economic sectors (industrial, energy, etc.) and all seasons by the same ratio. Such an adjustment significantly improved the simulated radiative forcing compared to the direct observations.



## 2.2 Simulations

This study used a series of simulations (Table 1):

**Fully Coupled CESM1 simulations**. The control case was a 394-year pre-industrial simulation (referred to as PI). Two perturbed simulations: sulfur dioxide ($SO_2$) (a precursor of $SO_4$) and BC emissions were increased instantaneously from pre-

industrial to present-day levels, but the GHG concentrations were unchanged (referred to as $PDSO_4$ and PDBC). Starting from the end of the 319th year, the perturbed simulations were run for 75 years, with the last 60 years being analyzed. To increase the signal-to-noise ratio caused by BC forcing (a smaller forcing), we performed an ensemble of five perturbed simulations. The long averaging time (394 years for the PI case, 60 years in the $SO_4$ perturbed simulations, and 300 years in the BC perturbed simulations) can restrain the impact of decadal natural climate variability and obtain a clear effect due to

aerosol forcings.

**Atmosphere-only model simulations with fixed SST**. The model settings were same as those in (1), but the SST was always fixed at the pre-industrial level, with only seasonal variability. The SST data is from the outputs of PI coupled simulation. Three simulations were performed: using the pre-industrial aerosol emissions (referred to as PI_FSST), present-day $SO_2$ emission (referred to as $PDSO_4$_FSST), and present-day BC emission (referred to as PDBC_FSST), respectively.

Each simulation was run for 75 years, with the last 60 years being analyzed. These three atmosphere-only simulations were also used to calculate the effective radiative forcings (ERFs) of $SO_4$ and BC at the present day following Myhre et al. (2013).

Those sets of simulations mentioned above have been adopted to examine the responses of the tropospheric atmosphere (Xu and Xie, 2015), mountain snow cover (Xu et al., 2016), and terrestrial aridity (Lin et al., 2016) to various forcing agents. The total response (TR) of the EASM to $SO_4$ or BC forcing was defined as the difference between $PDSO_4$ or PDBC and PI:

$$TRSO_4 = PDSO_4 - PI, \tag{1}$$

$$TRBC = PDBC - PI. \tag{2}$$

The fast response (FR) of the EASM to $SO_4$ or BC forcing was expressed as the difference between $PDSO_4$_FSST or PDBC_FSST and PI_FSST:

$$FRSO_4 = PDSO_4\_FSST - PI\_FSST, \tag{3}$$

$$FRBC = PDBC\_FSST - PI\_FSST. \tag{4}$$

Note that the slow response (SR) of the EASM to aerosol forcing, defined as climate response to aerosol-induced SST change, was calculated by subtracting the FR from the TR (Andrews et al., 2010; Ganguly et al., 2012; Samset et al., 2016), rather than by performing the simulations with perturbed SST pattern caused by aerosol forcing:

$$SR = TR - FR. \tag{5}$$

Hsieh et al. (2013) and Xu and Xie (2015) indicated that this approximate method was a legitimate metric to obtain the slow response of climate to aerosol forcing.



## 3 Results

### 3.1 Aerosol ERFs and their induced SST responses

Figure 1 shows the changes in $SO_4$ and BC optical depths at 550 nm from PI to PD. The aerosol optical depth increases significantly over most of the globe except for some oceans due to the increase in anthropogenic aerosol loading. The change

in $SO_4$ optical depth is larger than that in BC. The prominent increase in $SO_4$ optical depth appears over eastern China and USA, India, and Western Europe. The BC optical depth is increased dramatically over eastern China and South Asia.

The fifth assessment report of the Intergovernmental Panel on Climate Change (IPCC AR5) provided a new definition of radiative forcing named as ERF, which is a better indicator of the climate responses (Myhre et al., 2013). The global distributions of simulated $SO_4$ and BC ERFs at the top of the atmosphere (TOA) are shown in Figure 2. The ERFs are

calculated using the atmosphere-only model simulations with fixed SST by subtracting the net radiative flux at the TOA. There are fundamental differences between both aerosol ERFs. Reflecting $SO_4$ gives rise to large negative ERFs, especially in East and Southeast Asia, Central Africa, Western Europe, and subtropical oceans. However, absorbing BC leads to marked positive ERFs over East and South Asia and Central Africa, where the BC emission is large. The simulated global annual mean $SO_4$ and BC ERFs are -0.98 W m$^{-2}$ and +0.36 W m$^{-2}$, respectively. The simulated $SO_4$ forcing is close to those

estimated by Zelinka et al. (2014) and Forster et al. (2016), while our results show a larger BC forcing. This is attributed to the correction of BC emission in our simulations (Xu et al., 2013). The differences between reflecting and absorbing aerosol forcings imply the substantially different climate responses.

Aerosol-induced SST change is an important part of the climatic effect of aerosols (Xu and Xie, 2015). Figure 3 shows the changes in SST caused by various aerosol species from the fully coupled simulations. Despite the essential difference

between both types of forcings, the spatial pattern of SST change caused by $SO_4$ is found to be similar to that caused by BC (opposite in sign). It is characterized by a large SST change over the mid-latitude oceans of the Northern Hemisphere (NH), but only a slight SST change in the Southern Hemisphere (SH). Such an interhemispheric asymmetric adjustment in SST has been used as a crucial index of climate change (Ocko et al., 2014). The simulated global annual mean SST changes caused by $SO_4$ and BC are -0.44 K and +0.12 K, respectively.

### 3.2 Response of the EASM to $SO_4$ forcing

The sign of change in surface temperature is consistent well with that of the forcing. Negative $SO_4$ forcing leads to a marked surface cooling in summer over the East Asian monsoon region (EAMR), which increases with latitude (Figure 4a). In particular, the cooling exceeds 1 K over most of the NH subtropical oceans. The anomalous northerly winds prevail over eastern China and the surrounding oceans between 20$^o$N and 40$^o$N due to $SO_4$ forcing (Figure 4d), which signifies the

weakening of the EASM circulation. As seen in Figure 4, the slow responses of surface air temperature and winds at 850 hPa to $SO_4$-induced SST change close resembles the total responses of them to $SO_4$.



The fast response of surface air temperature to $SO_4$ forcing primarily features a cooling over the East Asian continent (Figure 4b) because the SST is fixed in these simulations and changes in $SO_2$ emissions are concentrated over land. Such a change in surface temperature decreases the land-sea surface thermal contrast over the EAMR, thus weakening the EASM circulation (Figure 4e). This is consistent with previous studies using other general circulation models with fixed SST (e.g., Jiang et al., 2013; Dong et al., 2016). However, note that the weakening of the EASM in fast response to $SO_4$ is too weak to explain the total response of the EASM to $SO_4$, especially over eastern China (Figure 4d and e). Therefore, we next elaborate the physical mechanism behind the slow response of the EASM to $SO_4$.

Figure 5 shows the JJA mean responses of zonally averaged atmospheric temperature between 100°E and 140°E to $SO_4$ forcing over the EAMR. The $SO_4$-induced slow climate response leads to a significant cooling in the whole troposphere (Figure 5c), though $SO_4$ doesn't largely affect the radiation in the atmosphere. It is responsible for a large fraction of the atmospheric cooling in total response to $SO_4$ (Figure 5a and c). This is because the interhemispheric asymmetric change in SST caused by $SO_4$ may distinctly affect the free troposphere by alerting the tropical circulations and mid-latitude eddies (Hsieh et al., 2013; Ocko et al., 2014; Xu and Xie, 2015). The most remarkable feature of change in atmospheric temperature in slow response to $SO_4$ is a deep tropospheric cooling between 30°N and 45°N (Figure 5c). A similar temperature response to aerosol forcings was found by Rotstayn et al. (2014) based on the multi-model ensemble simulations, which indicates that this is a robust feature of climate response to aerosols. There is an anomalous cooling center at the upper troposphere (200 – 500 hPa) (Figure 5c), which leads to a prominent decrease in geopotential height at those altitudes and drop in pressure at about 200 hPa (Figure 5f).

The East Asian subtropical jet (EASJ) that is located around 40°N at 200 hPa is an important component of the East Asian monsoon. The pressure drop at 200 hPa in slow response to $SO_4$ increases the poleward (equatorward) pressure gradient force to the south (north) of the cooling region. Such a change in pressure gradient force leads to increase (decrease) in westerlies to the south (north) of the EASJ center through the geostrophic balance between the Coriolis force and pressure gradient force (Yu et al., 2004). It is shown in Figure 6a that the largest increase and decrease in westerlies occur at about 25°N and 45°N, respectively. Consequently, the EASJ shifts southwards in response to $SO_4$. The slow response dominates over the total response of the EASJ to $SO_4$ (Figure 6a and c). According to the thermal wind principle, the southward displacement of the EASJ will result in an anomalous anticyclone over the East Asian continent (Zhu et al., 2012). To the east of the anticyclonic center, anomalous northerlies increase prominently (Figure 4f). In addition, the interhemispheric SST gradient caused by $SO_4$ forcing (Figure 3a) strengthens the ascending branch of the local Hadley cell in the summer (Figure not shown), thereby resulting in an anomalous cyclonic vortex over southeastern China and the western Pacific between 15°N and 35°N (Figure 4f). To the west of the cyclonic center, anomalous northerly winds are further increased. Finally, the $SO_4$-induced slow climate response leads to a more intense weakening of the EASM circulation than its fast response. Dai et al. (2013) also suggested that the thermal contrast in the mid-upper troposphere played a more important role than that in the mid-lower troposphere in impacting the strength and variations of the Asian summer monsoon circulations. However, the





changes in atmospheric temperature and geopotential height due to the adjustments in clouds and atmospheric states in the fast response to $SO_4$ (Figures 5b and e) lead to the increase in westerlies at the north of the jet center (Figures 6b). Note that the positive change of westerlies in fast response is comparable to the negative change of westerlies in slow response to $SO_4$ due to the comparable changes in temperature and geopotential height. The change of the jet in fast response to $SO_4$ is conducive to the enhancement of the EASM (Figure 6b), which partially offsets the weakening of the EASM due to the decrease of land-sea surface thermal contrast.

The weakening of the EASM circulation caused by $SO_4$ forcing suppresses the transport of surface warm and moist air northwards and upwards, which results in significant decrease in precipitation over eastern and southern China and the ambient oceans (Figure 7a). However, the precipitation increases (yet not significantly) over some of the western Pacific due to the enhanced convection in slow response to $SO_4$. The $SO_4$-induced slow climate response leads to a larger and even opposite change in precipitation over some of the EAMR compared to its fast response (Figure 7b and c). This indicates the importance of ocean response to $SO_4$ forcing in driving the changes of the EASM.

### 3.3 Response of the EASM to BC forcing

Figure 8 shows JJA mean responses of surface air temperature and wind vectors at 850 hPa to BC forcing. Absorbing BC increases the surface air temperature over the EAMR, with the largest warming appearing at the NH mid-latitudes, especially over the northwestern Pacific (Figure 8a). This is mainly from the contribution of slow climate response to BC (Figure 8c). The significant anomalous southerly winds at 850 hPa prevail, and the EASM circulation is enhanced north of 30°N in total climate response to BC (Figure 8d), which is mainly attributed to their fast responses to BC. The fast and slow responses of surface winds to BC are different south of 30°N. The anomalous northerlies in slow response that tend to weaken the EASM are slightly stronger than the anomalous southerlies in fast response to BC (Figure 8f and e). Finally, the total climate response caused by BC weakened the EASM circulation slightly south of 30°N (Figure 8d).

Now we explain why the enhancement of monsoon in fast response to BC forcing is strong. Firstly, the large surface warming in fast response to BC occurs over the East Asian continent, especially north of 30°N (Figure 8b). This increases the land-sea surface thermal contrast over the EAMR, thereby enhancing the EASM circulation (Figure 8e). This mechanism is also at work in fast response to $SO_4$. Secondly and unique to the BC case, the direct absorption of solar radiation by BC leads to a deep tropospheric warming at the NH mid-latitudes (Figure 9), which dominates over the tropospheric warming in total response to BC. An anomalous warming center appears and the geopotential height increases at the upper troposphere of around 40°N (Figure 9e). Consequently, the pressure increases at the uppermost troposphere, which strengthens the poleward (equatorward) pressure gradient force to the north (south) side of warming region. This results in an increase (decrease) in westerly winds to the north (south) side of the EASJ center and the northward movement of the EASJ (Figure 10b). The total response of the jet is consistent with the fast response of it to BC. With the change of the EASJ, an anomalous cyclonic vortex is formed over the East Asian land, and anomalous southerly winds increase over eastern China.





This second mechanism involving the EASJ change further magnifies the enhancement of the EASM caused by the increase in land-sea surface thermal contrast in fast response to BC.

BC-induced slow response is in the opposite direction of the fast response. As a result of the marked increase in SST over the NH (Figure 3b), the BC-induced slow climate response also strengthens the ascending branch of the local Hadley cell during the summer (Figure not shown). This leads to an anomalous cyclone in the lower atmosphere between 15°N and 30°N, thus increasing the anomalous northerly winds over eastern China (Figure 8f). While the tropospheric temperature increases in slow response to BC, the warming in the upper troposphere of around 40°N is less than those on both sides of it (Figure 9c). Such an adjustment in tropospheric temperature is conducive to a southward shifting of the EASJ (Figure 10c). These EASJ changes cause the BC-induced slow response to weaken the EASM circulation, which even overcomes the strengthening of the EASM due to the increase in land-sea surface thermal contrast in slow response (Figure 8c).

Lastly, the JJA mean response of precipitation to BC forcing over the EAMR is weaker than those found in response to SO4 with less area with significant changes (Figure 11), mainly because of a smaller radiative forcing. The total response of precipitation to BC manifests a spatial pattern of "wetting-drying-wetting" from north to south over the EAMR (Figure 11a). This is not consistent with that reported by Menon et al. (2002), which indicated that BC forcing primarily contributed to the "wetter-south-dryer-north" phenomenon in eastern China during the past decades. The change in precipitation caused by BC forcing is mainly in line with the change in monsoon circulation. The fast and slow responses of precipitation to BC are almost opposite over the EAMR (Figure 11b and c) due to the opposite circulation changes. The spatial distribution of total precipitation response agrees well with that of slow precipitation response to BC, which also shows the significance of SST change induced by BC forcing in impacting the EASM.

## 4 Discussion and conclusions

This study investigates the roles of fast and slow components in shaping the total equilibrium response of the EASM to reflecting $SO_4$ and absorbing BC forcings using an Earth system model with a fully coupled dynamic ocean, in contrast to most of the previous studies that adopted a slab ocean model (e.g., Allen and Sherwood, 2010; Ganguly et al., 2012). Such a decomposition of total response will be helpful to better understanding the mechanisms by which aerosols impact the EASM. Our results show that reflecting $SO_4$ produces a global mean ERF of -0.98 W m$^{-2}$ while absorbing BC leads to an ERF of +0.36 W m$^{-2}$. Despite the essential difference in forcings, the spatial distribution of SST response at the global scale is prominently similar between $SO_4$ and BC forcings.

There are significantly different mechanisms between fast and slow responses of the EASM to different aerosol forcings. Table 2 provides a summary of the responses of the EASM in various cases. The $SO_4$-induced fast climate response weakens the EASM through decreasing the land-sea surface thermal contrast over the EAMR. This has been shown in many earlier studies (e.g., Jiang et al., 2013; Wang et al., 2015). However, we show here that the $SO_4$-induced SST change (i.e., slow





climate response) further weakens the EASM by changing the tropospheric thermodynamic and circulation structures, especially through a southward shifting of the EASJ. Eventually, the EASM circulation is significantly weakened, and the precipitation is reduced over the EAMR in the total response to $SO_4$. We emphasize that the $SO_4$-induced slow response plays a more important role in driving the changes of the EASM.

The BC-induced changes are weaker and more complicated. The fast climate response significantly strengthens the EASM both by increasing the land-sea surface thermal contrast over the EAMR and moving the EASJ northwards. However, the BC-induced slow climate response weakens the EASM through strongly affecting the atmospheric temperature and circulation. The role of the EASJ hasn't been clearly shown in previous studies that often only considered the fast adjustment of climate to BC forcing. As a result of the competing factors of land-sea contrast and ESAJ shift, the EASM in the total

response to BC is weaker and less significant, with an enhancement north of 30°N (northern China), but a slightly weakening south of 30°N (southern China). As for the precipitation responses, the total response to BC shows a spatial pattern of "wetting-drying-wetting" from north to south over the EAMR. This differs from the results in Menon et al. (2002), which suggested that the increased BC emission contributed to the "wetter-south-dryer-north" phenomenon in summer over eastern China in the past decades.

This study elaborates the mechanisms of the impacts of various aerosol species on the EASM system, highlighting the importance of ocean response to aerosol forcings (i.e., slow response component) in driving the changes of the EASM. Given a larger negative ERF due to SO4, it can be speculated that the integrated effect of total anthropogenic aerosols likely tends to weaken the EASM over the industrial era, as suggested by earlier works (e.g., Song et al., 2014; Salzmann et al., 2014).

Our results clearly suggest that one pathway for aerosol forcings to affect the EASM is by changing the land-sea surface thermal contrast, as shown in previous studies (e.g., Liu et al., 2011; Zhang et al., 2012; Salzmann et al., 2014; Wang et al., 2015). But we also emphasize the role of the EASJ, which could amplify or offset the effects of surface thermal contract. The response of the EASJ to aerosols need further studies preferably using multi-model ensembles, because (1) it is quite sensitive to the atmospheric forcing component (Figure 10b) that is altitude dependent and (2) as a component of the larger

NH westerly jet stream, it is more subject to the influence of non-local (outside Asia) aerosols that could undergo a different emission pathway than local aerosol emissions in a shorter time.

**Data availability**

The model outputs analyzed in this study can be accessed by contacting the corresponding author Lei Lin

(linlei3@mail.sysu.edu.cn).

**Author contributions**



Z. Wang and L. Lin conceived the study and performed the analysis. Z. Wang, L. Lin, M. Yang, and Y. Xu wrote the paper. All authors provided comments and contributed to the text.

**Competing interests**

The authors declare that they have no conflict of interest.

**Acknowledgments**

This study was supported by the (key) National Natural Science Foundation of China (41575139 and 91644211) and National Key Project of MOST (2016YFC0203306).

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



**Table 1: Simulation setups.**

| Simulation | Aerosol emissions | Ocean | Ensembles |
| --- | --- | --- | --- |
| PI | Year 1850 $SO_2$ and BC | Dynamic ocean model | 1 |
| PDSO$_4$ | Year 2000 $SO_2$ and 1850 BC | Dynamic ocean model | 1 |
| PDBC | Year 1850 $SO_2$ and 2000 BC | Dynamic ocean model | 5 |
| PI_FSST | Year 1850 $SO_2$ and BC | Fixed SST from PI | 1 |
| PDSO$_4$_FSST | Year 2000 $SO_2$ and 1850 BC | Fixed SST from PI | 1 |
| PDBC_FSST | Year 1850 $SO_2$ and 2000 BC | Fixed SST from PI | 1 |





**Table 2: Summary of the fast and slow responses of the EASM to SO$_4$ and BC forcings.**

| | Fast response | | | | Slow response | | | |
|---|---|---|---|---|---|---|---|---|
| | Thermal contrast | Subtropical jet | Southerly winds | Precipitation | Thermal contrast | Subtropical jet | Southerly winds | Precipitation |
| SO$_4$ | Decrease | Northward shift (weakly) | Decrease (weakly) | Decrease | Decrease (weakly) | Southward shift | Decrease | Decrease |
| BC | Increase | Northward shift | Increase | Decrease | Increase (weakly) | Southward shift (weakly) | Decrease | Increase (weakly) |



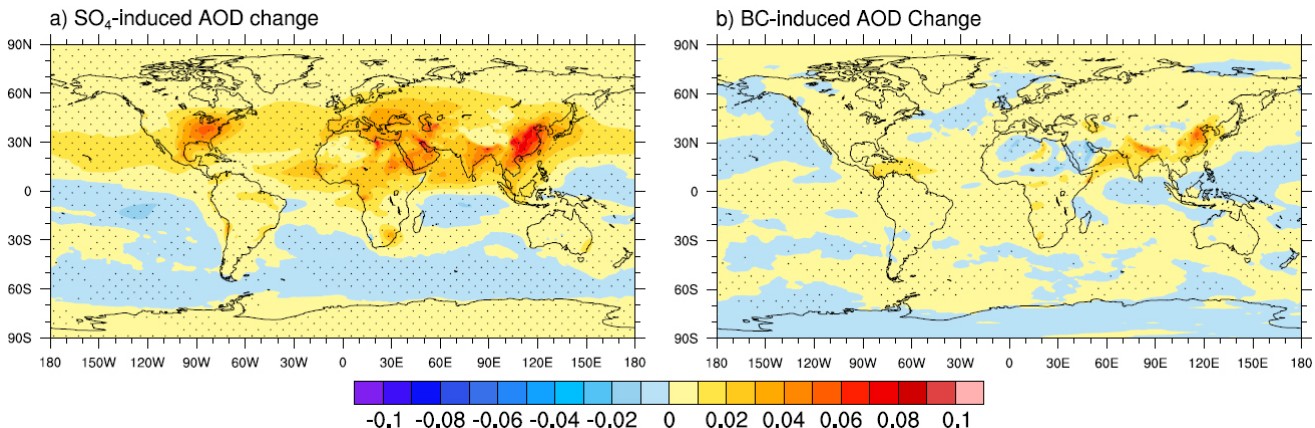

**Figure 1: Annual mean distributions of changes in (a) SO$_4$ and (b) BC optical depths at 550 nm from PI to PD.**





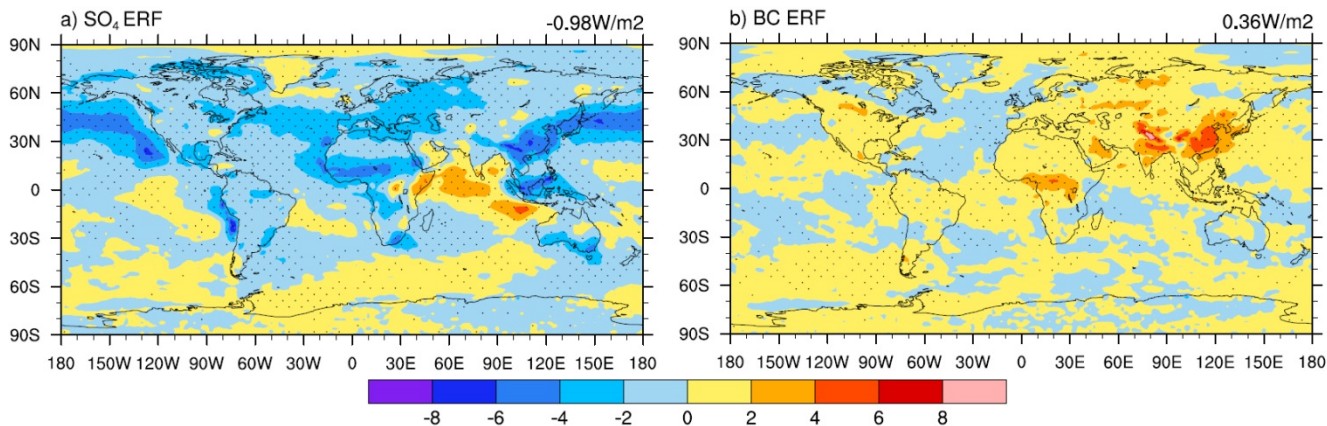

**Figure 2: Annual mean distributions of (a) SO$_4$ and (b) BC ERF from PI to PD (unit: W m$^{-2}$). ERF is defined as the perturbation of net radiative flux at the TOA caused by aerosols.**





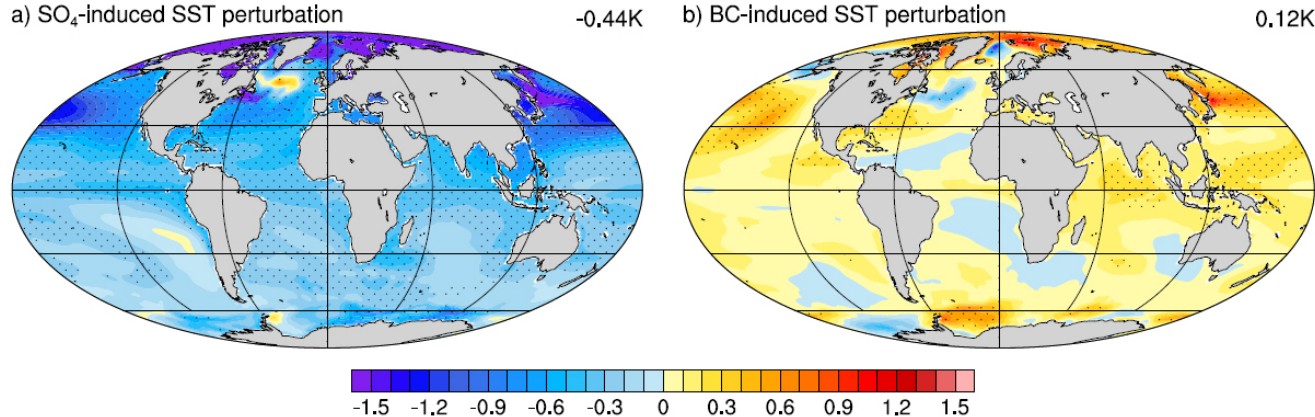

**Figure 3: Annual mean distributions of SST responses to (a) SO$_4$ and (b) BC forcings (units: K).**





**Figure 4: JJA mean total, fast, and slow responses of (a, b, c) surface air temperature (unit: K) and (d, e, f) wind vectors at 850 hPa (unit: m s⁻¹) to SO₄ forcing.**





**Figure 5: JJA mean total, fast, and slow responses of zonally averaged (a, b, c) atmospheric temperature (unit: K) and (d, e, f) geopotential height (unit: m) between 100°E and 140°E to SO$_4$ forcing.**





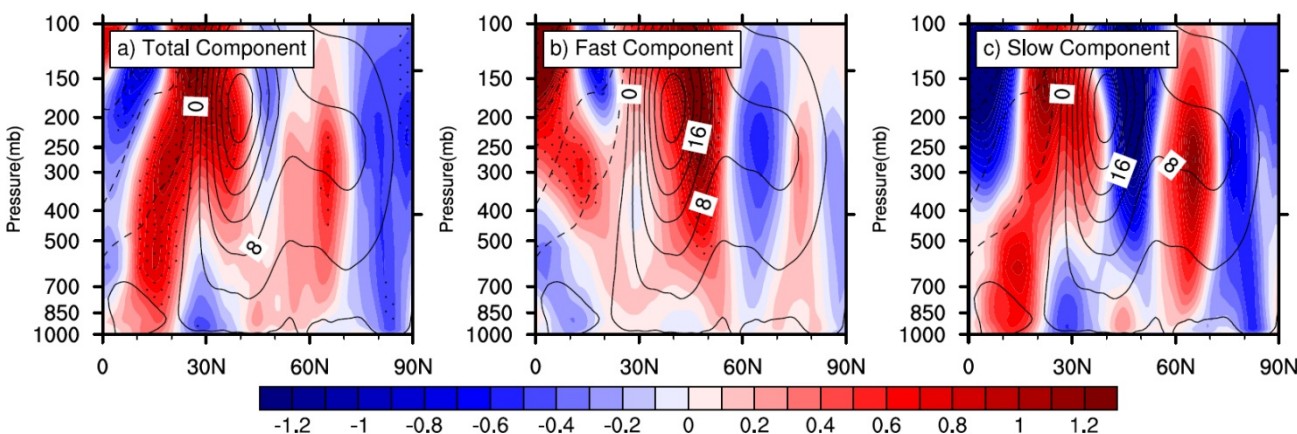

**Figure 6: JJA mean total, fast, and slow responses of zonally averaged zonal wind between 100°E and 140°E to SO$_4$ forcing (unit: m s$^{-1}$). The dashed and solid lines represent the climatological JJA mean easterly and westerly winds in PI, respectively.**





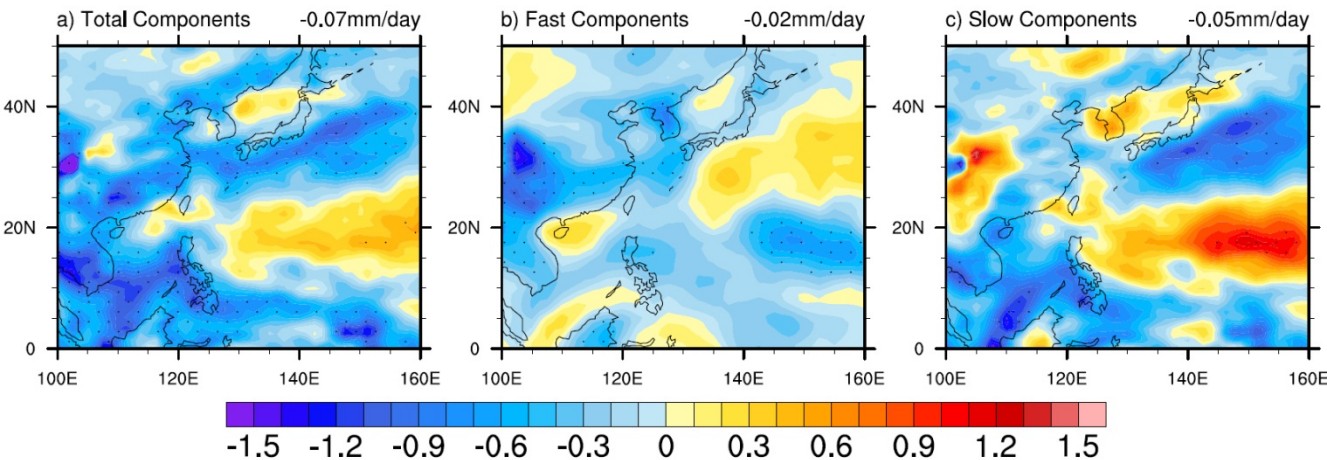

**Figure 7: JJA mean total, fast, and slow responses of precipitation rate to SO$_4$ forcing (unit: mm day$^{-1}$). The values in the top right corner of the figures represent the responses averaged over the region 0°-50°N, 100°-140°E.**







**Figure 8: JJA mean total, fast, and slow responses of (a, b, c) surface air temperature (unit: K) and (d, e, f) wind vectors at 850 hPa (unit: m s⁻¹) to BC forcing.**




**Figure 9: JJA mean total, fast, and slow responses of zonally averaged (a, b, c) atmospheric temperature (unit: K) and (d, e, f) geopotential height (unit: m) between 100°E and 140°E to BC forcing.**





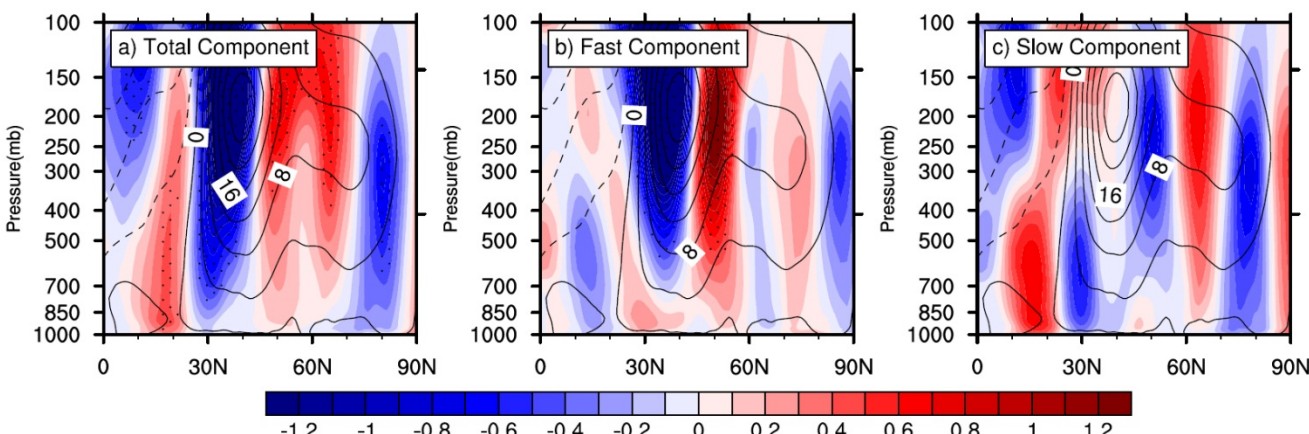

**Figure 10: JJA mean total, fast, and slow responses of zonally averaged zonal wind between 100°E and 140°E to BC forcing (unit: m s$^{-1}$). The dashed and solid lines represent the climatological JJA mean easterly and westerly winds in PI, respectively.**

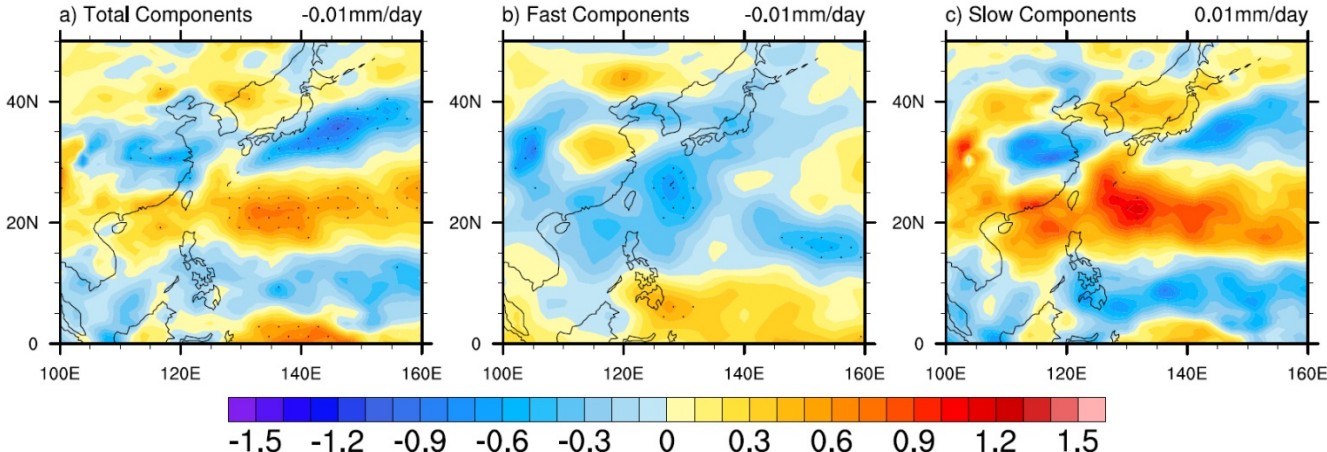

Figure 11: JJA mean total, fast, and slow responses of precipitation rate to BC forcing (unit: mm day$^{-1}$). The values in the top right corner of the figures represent the responses averaged over the region 0°-50°N, 100°-140°E.