# Peer review of "Disentangling fast and slow responses of the East Asian summer monsoon to reflecting and absorbing aerosol forcings"

_Atmospheric Chemistry and Physics, 2017_

## Referee Comment (RC1) · Anonymous Referee #2 · 23 Jun 2017

Reviewer's comments for the paper (ACP-2017-464), entitled "Disentangling fast and slow responses of the East Asian summer monsoon to reflecting and absorbing aerosol forcings" by Wang et al., submitted to ACP

Recommendation, Major revision

General comments By performing some time slice experiments using an AGCM and a coupled GCM, this paper investigates the fact and slow responses of East Asian summer monsoon to changes of sulphur dioxide (SO2) and black carbon (BC) from preindustrial to present day. While the topic is an interesting one. However, results in current version of paper is not very well presented and some conclusions are lack

of evidence to support them. Therefore, paper needs a major revision by addressing some major and specific comments listed below before it can be accepted for publication. Major comment 1. Some conclusions for precipitation changes over East Asia are based on areal average over a large domain including both land and ocean. The precipitation responses to different forcings show some clear contrast features over land over East Asia and adjacent ocean. For example, Fig. 7 shows that the decrease of precipitation over land over East Asia in total response to SO2 change is dominated by the fast response while changes over adjacent ocean might be dominated by slow response. Therefore, some statements about fast response and slow response of EASM to the SO2 change are misleading by using large areal average. 2. The current version of the paper lacks quantitative statements when either the fast or slow responses to different forcings are described. This aspect needs to be improved. 3. It is worth of discussing the JJA SST responses since the paper is about EASM. 1. There are detailed analyses of zonal averaged temperature and zonal wind over sector 100E-140E in response to different forcings. However, the relation between these sectorial averaged changes with the regional pattern of precipitation changes is not clearly illustrated. Specific comments 1. Lines 24-25 on page 1. "Consequently, the EASM is enhanced north of 30°N but slightly reduced south of 30°N in the total response to BC.". This statement is very confusing. EASM is a summer climate system over East Asia. It is difficulty to follow your argument that EASM north of 30N enhances and south of 30N weakens. 2. In several places, it states "pressure drop at 200 hPa" or "drop in pressure at 200 hPa". This does not make sense. 3. Lines 25-27 on page 6. How does "the southward displacement of the EASJ will result in an anomalous anticyclone over the East Asian continent"? Need some explanations. 4. Lines 27-28. It is not clear which season "there is an enhanced NH Hadley cell" since figure 3 is annual SST responses. 5. Lines 5-6 on page 8. "This leads to increase in vertical ascending motion between 20°N and 40°N (the position of subsiding branch of the NH Hadley cell)". This is confusing. Subsiding branch of local Hadley Cell shall be in southern hemisphere in JJA. 6. Lines 4-5 on page 9. "We emphasize that the SO4-induced

slow response plays a more important role in driving the changes of the EASM." See major comment 1. 7. Lines 10-11 on page 9. "the EASM in the total response to BC is weaker and less significant, with an enhancement north of 30°N (northern China), but a slightly weakening south of 30°N (southern China).". see specific comment 1.

---

## Referee Comment (RC2) · Anonymous Referee #1 · 5 Jul 2017

This paper elaborates the role of fast-response and slow-response in shaping the total equilibrium response of the East Asian summer monsoon (EASM) to SO4 and BC using the Community Earth System Model. This paper states the importance of ocean response to aerosol forcing in driving the changes of the EASM. I recommend for its publication after revision.

Comments: (1) Lack of clarity at a few places in the manuscript. For example: In section 2.2 add details of how an ensemble of five perturbed simulations is obtained. Explain why the averaging time of SO4 (60 yrs) is less than BC (300 yrs).

Section 3

[Figure]

Line 1: Do you mean changes in optical depth induced by SO4 or BC?

Lines 4-5: "The aerosol optical depth increases significantly over most of the globe except for some oceans due to the increase in anthropogenic aerosol loading". Mention the region of the oceans.

Lines 22-23: Comparison with previous studies if any.

Figure 3: It is interesting to show opposite response due to SO4 and BC aerosols on SSTs. This discussion should be elaborated.

It will also be interesting to know the simulated annual mean SST changes caused by SO4 and BC for the northern and southern hemisphere separately.

Anomalies in wind should be plotted in Figure 5a-c. It will provide information on circulation changes.

I suggest plotting tropopause anomalies in Figures 5d-f. Drop in geopotential height and tropopause has linkages with suppression on monsoon convection and weakening of the EASM.

I also suggest plotting monsoon Hadley circulations in Figure 7 and 11 along with precipitation. Discussions on subsidence/ascending motion will be helpful.

Add confidence level (95% or 99%) in figures 7 and 11 as hitch lines to show the results are significant.

I think that Figures 7 and 11, both indicate that for BC and SO4 (both) precipitation pattern for the total-response is a sum of the fast-response and the slow-response. If the magnitude of the fast-response is higher than the slow-response, then it dominates. Please verify by quantitative analysis and results should be modified accordingly.

Table-2 suggest that BC induces a weak increase in precipitation due to slow response and decrease due to fast response. While SO4 induces a decrease in precipitation due to both slow and fast response. Figure 7 and 11 suggest that total response induced

by SO4 weakens the EASM but total response by BC aerosols is "wetter-south-dryer-north". Do this wetter-south-dryer-north points to weak increase or decrease in overall precipitation? Does this study conclude that SO4 and BC both cause a decrease in overall precipitation?

———————————————————

---

## Author Comment (AC1) · 31 Jul 2017

**Response to Referee #2:**

**We first thank the helpful comments of the reviewer. We have taken reviewer's comments into consideration and revised the manuscript accordingly. All the changes have been highlighted in the revised manuscript. Our detailed responses are as follows.**

*Reviewer's comments for the paper (ACP-2017-464), entitled "Disentangling fast and slow responses of the East Asian summer monsoon to reflecting and absorbing aerosol forcings" by Wang et al., submitted to ACP.*

*Recommendation, Major revision.*

*General comments*

*By performing some time slice experiments using an AGCM and a coupled GCM, this paper investigates the fact and slow responses of East Asian summer monsoon to changes of sulphur dioxide ($SO_2$) and black carbon (BC) from preindustrial to present day. While the topic is an interesting one. However, results in current version of paper are not very well presented and some conclusions are lack of evidence to support them. Therefore, paper needs a major revision by addressing some major and specific comments listed below before it can be accepted for publication.*

**Response: We have addressed all the comments and revised the manuscript. Please see the specific description below.**

*Major comment*

*1. Some conclusions for precipitation changes over East Asia are based on areal average over a large domain including both land and ocean. The precipitation responses to different forcings show some clear contrast features over land over East*

5 *Asia and adjacent ocean. For example, Fig. 7 shows that the decrease of precipitation over land over East Asia in total response to $SO_4$ change is dominated by the fast response while changes over adjacent ocean might be dominated by slow response. Therefore, some statements about fast response and slow response of EASM to the $SO_4$ change are misleading by using large areal average.*

10 **Response: Thanks for the reviewer's comment. We have improved these statements. Please see the last paragraph in page 7, the first paragraph in page 8, and line 25 – 31 in page 9 in the revised manuscript.**

*2. The current version of the paper lacks quantitative statements when either the fast*

15 *or slow responses to different forcings are described. This aspect needs to be improved.*

**Response: Accepted. We have improved this aspect. Please see the revised manuscript.**

20 *3. It is worth of discussing the JJA SST responses since the paper is about EASM.*

**Response: Accepted. We have added the figures of JJA SST responses in the supplement material and the corresponding discussions. Please see the Figure S1,**

**line 21 – 23 in page 6, and line 3 – 5 in page 9 in the revised manuscript.**

*4. There are detailed analyses of zonal averaged temperature and zonal wind over sector $100^oE$-$140^oE$ in response to different forcings. However, the relation between these sectorial averaged changes with the regional pattern of precipitation changes is not clearly illustrated.*

**Response: Accepted. We have added the illustration about the relation between the changes in zonal averaged temperature and zonal wind with the regional pattern of precipitation changes. Please see the last paragraph in page 7, the first paragraph in page 8, and line 25 – 31 in page 9 in the revised manuscript.**

*Specific comments*

*1. Lines 24-25 on page 1. "Consequently, the EASM is enhanced north of $30^oN$ but slightly reduced south of $30^oN$ in the total response to BC.". This statement is very confusing. EASM is a summer climate system over East Asia. It is difficulty to follow your argument that EASM north of $30^oN$ enhances and south of $30^oN$ weakens.*

**Response: We have corrected this sentence in the revised manuscript.**

*2. In several places, it states "pressure drop at 200 hPa" or "drop in pressure at 200 hPa". This does not make sense.*

**Response: We have removed it in the revised manuscript.**

*3. Lines 25-27 on page 6. How does "the southward displacement of the EASJ will result in an anomalous anticyclone over the East Asian continent"? Need some explanations.*

**Response: Accepted. We have added the explanations. Please see the line 6 – 8 in page 7 in the revised manuscript.**

*4. Lines 27-28. It is not clear which season "there is an enhanced NH Hadley cell" since figure 3 is annual SST responses.*

**Response: Here is the response of Hadley cell in the summer. We have improved this statement. Please see the line 11 – 12 in page 7 in the revised manuscript.**

*5. Lines 5-6 on page 8. "This leads to increase in vertical ascending motion between $20^oN$ and $40^oN$ (the position of subsiding branch of the NH Hadley cell)". This is confusing. Subsiding branch of local Hadley Cell shall be in southern hemisphere in JJA.*

**Response: Yes, it should be the ascending branch of local Hadley Cell. We have corrected it. Please see the line 9 – 10 in page 9 in the revised manuscript.**

*6. Lines 4-5 on page 9. "We emphasize that the $SO_4$-induced slow response plays a more important role in driving the changes of the EASM." See major comment 1.*

**Response: This has been revised. Please see the line 16 – 18 in page 10 in the revised manuscript.**

*7. Lines 10-11 on page 9. "the EASM in the total response to BC is weaker and less significant, with an enhancement north of $30^oN$ (northern China), but a slightly weakening south of $30^oN$ (southern China).". see specific comment 1.*

**Response: This has been revised. Please see the line 23 – 24 in page 10 in the revised manuscript.**

---

## Author Comment (AC2) · 31 Jul 2017

**Response to Referee #1:**

**We first thank the valuable comments of the reviewer. We have taken reviewer's comments into consideration and revised the manuscript accordingly. All the changes have been highlighted in the revised manuscript. Our detailed responses are as follows.**

*This paper elaborates the role of fast-response and slow-response in shaping the total equilibrium response of the East Asian summer monsoon (EASM) to $SO_4$ and BC using the Community Earth System Model. This paper states the importance of ocean response to aerosol forcing in driving the changes of the EASM. I recommend for its publication after revision.*

*Comments: (1) Lack of clarity at a few places in the manuscript. For example: In section 2.2 add details of how an ensemble of five perturbed simulations is obtained. Explain why the averaging time of $SO_4$ (60 yrs) is less than BC (300 yrs).*

**Response: The ensemble of five perturbed simulations for BC was performed by altering the atmospheric initial conditions by an air temperature difference at round-off level (order of $10^{-14}$ ℃). To enhance the signal-to-noise ratio of the response, the 300 years in the BC perturbed simulations (60 years × 5 members) was conducted because of weaker BC forcing than $SO_4$. We have added these statements. Please see line 8 – 9 in page 4 in the revised manuscript.**

*Section 3*

*Line 1: Do you mean changes in optical depth induced by $SO_4$ or BC?*

**Response: Yes, it is. We have corrected it. Please see line 3 in page 5 in the revised manuscript.**

*Lines 4-5: "The aerosol optical depth increases significantly over most of the globe except for some oceans due to the increase in anthropogenic aerosol loading". Mention the region of the oceans.*

**Response: Done. Please see line 5 – 9 in page 5 in the revised manuscript.**

10

*Lines 22-23: Comparison with previous studies if any.*

**Response: Done. Please see line 28 – 31 in page 5 in the revised manuscript.**

*Figure 3: It is interesting to show opposite response due to $SO_4$ and BC aerosols on*

15 *SSTs. This discussion should be elaborated.*

**Response: Accepted. We have elaborated the opposite SST responses due to $SO_4$ and BC. Please see line 25 – 28 in page 5 in the revised manuscript.**

*It will also be interesting to know the simulated annual mean SST changes caused by*

20 *$SO_4$ and BC for the northern and southern hemisphere separately.*

**Response: Accepted. We have added the SST changes caused by $SO_4$ and BC for both hemispheres separately. Please see the line 31 in page 5 – line 1 in page 6**

**and Table 2 in the revised manuscript.**

*Anomalies in wind should be plotted in Figure 5a-c. It will provide information on circulation changes.*

5 **Response: Accepted. We have added the figures of anomalous winds induced by SO$_4$ and BC in the supplement material. Please see the Figures S2 and S4.**

*I suggest plotting tropopause anomalies in Figures 5d-f. Drop in geopotential height and tropopause has linkages with suppression on monsoon convection and weakening*

10 *of the EASM.*

**Response: Accepted. We have added the figures of tropopause height anomalies induced by SO$_4$ and BC and the corresponding discussions. Please see the Figure 7, line 17 – 21 in page 7, line 31 in page 8 – line 2 in page 9, and line 15 – 16 in page 9 in the revised manuscript.**

15

*I also suggest plotting monsoon Hadley circulations in Figure 7 and 11 along with precipitation. Discussions on subsidence/ascending motion will be helpful.*

**Response: Accepted. We have added the figures of changes in meridional circulations and stream functions induced by SO$_4$ and BC in the supplement**

20 **material and the corresponding discussions. Please see the Figure S2 – S5, line 32 – 33 in page 7, and line 25 – 29 in page 9 in the revised manuscript.**

*Add confidence level (95% or 99%) in figures 7 and 11 as hitch lines to show the results are significant.*

**Response: This has been added. Please see the revised figures.**

5  *I think that Figures 7 and 11, both indicate that for BC and SO$_4$ (both) precipitation pattern for the total-response is a sum of the fast-response and the slow-response. If the magnitude of the fast-response is higher than the slow-response, then it dominates. Please verify by quantitative analysis and results should be modified accordingly.*

**Response: Accepted. We have modified the statements about which response**

10  **dominates over the total response of precipitation to BC and SO$_4$ in different regions. Please see the line 2 – 8 in page 8 and line 25 – 31 in page 9 in the revised manuscript.**

*Table-2 suggests that BC induces a weak increase in precipitation due to slow*

15  *response and decrease due to fast response. While SO$_4$ induces a decrease in precipitation due to both slow and fast response. Figure 7 and 11 suggest that total response induced by SO$_4$ weakens the EASM but total response by BC aerosols is "wetter-south-dryer-north". Do this wetter-south-dryer-north points to weak increase or decrease in overall precipitation? Does this study conclude that SO$_4$ and BC both*

20  *cause a decrease in overall precipitation?*

**Response: Table 2 was aimed at the regional mean changes over East Asia. It could be misleading by using large areal average. Thus, we improved the**

**statements in this Table. Please see the Table 3 in the revised manuscript.**